# *TBX5* R264K acts as a modifier to develop dilated cardiomyopathy in mice independently of T-box pathway

**Nariaki Miyao**[1]ʘ, **Yukiko Hata**[2], **Hironori Izumi**[3], **Ryo Nagaoka**[4], **Yuko Oku**[2], **Ichiro Takasaki**[5], **Taisuke Ishikawa**[6], **Shinya Takarada**[1], **Mako Okabe**[1], **Hideyuki Nakaoka**[1], **Keijiro Ibuki**[1], **Sayaka Ozawa**[1], **Tomoyuki Yoshida**[3], **Hideyuki Hasegawa**[4], **Naomasa Makita**[6], **Naoki Nishida**[2], **Hisashi Mori**[3], **Fukiko Ichida**[1], **Keiichi Hirono**[1]ʘ*

1 Department of Pediatrics, University of Toyama, Toyama, Japan, 2 Department of Legal Medicine, University of Toyama, Toyama, Japan, 3 Department of Molecular Neuroscience, University of Toyama, Toyama, Japan, 4 Laboratory of Medical Information Sensing, University of Toyama, Toyama, Japan, 5 Department of Pharmacology, Graduate School of Science and Engineering, University of Toyama, Toyama, Japan, 6 Omics Research Center, National Cerebral and Cardiovascular Center, Osaka, Japan

ʘ These authors contributed equally to this work.
* khirono1973@gmail.com

**Data Availability Statement:** All relevant data are within the manuscript and its Supporting Information files.

## Abstract

### Background

TBX5 is a transcription factor that has an important role in development of heart. *TBX5* variants in the region encoding the T-box domain have been shown to cause cardiac defects, such as atrial septal defect or ventricular septal defect, while *TBX5* variants have also been identified in a few cardiomyopathy patients and considered causative. We identified a *TBX5* variant (c.791G>A, p.Arg264Lys), that is over-represented in cardiomyopathy patients. This variant is located outside of the T-box domain, and its pathogenicity has not been confirmed by functional analyses.

### Objective

To investigate whether the *TBX5* R264K is deleterious and could contribute to the pathogenesis of cardiomyopathy.

### Methods and results

We developed mice expressing Tbx5 R264K. Mice homozygous for this variant displayed compensated dilated cardiomyopathy; mild decreased fractional shortening, dilatation of the left ventricle, left ventricular wall thinning and increased heart weight without major heart structural disorders. There was no difference in activation of the *ANF* promotor, a transcriptional target of *Tbx5*, compared to wild-type. However, analysis of RNA isolated from left ventricular samples showed significant increases in the expression of *Acta1* in left ventricle with concomitant increases in the protein level of ACTA1.

**Funding:** The authors received no specific funding for this work.

**Competing interests:** The authors have declared that no competing interests exist.

**Abbreviations:** A, trans-mitral atrial wave; ACTA1, actin, alpha 1, skeletal muscle; ANF, Atrial natriuretic factor; ANKRD1, ankyrin repeat domain 1; ANKRD23, ankyrin repeat domain 23; ANOVA, analysis of variance; ASD, atrial septal defect; AWD, anterior wall thickness at end diastole; BRG1, Brahma-related gene-1; BW, body weight; DCM, dilated cardiomyopathy; E, trans-mitral early wave; FS, fractional shortening; GATA4, GATA binding protein 4; gnomAD, Genome Aggregation Database; HW, heart weight; LMNA, lamin A/C; LVDd, left ventricle diameter at end diastole; LVDs, left ventricle diameter at end systole; LVNC, left ventricular non compaction; MYBPC, myosin binding protein C; MYH, myosin heavy chain; MYL, myosin light chain; MYOT, myotilin; NGS, next generation sequencing; Nkx2.5, NK2 homeobox 5; NPPA, natriuretic peptide type A; NuRD, nucleosome remodeling and deacetylase; PWD, posterior wall thickness at end diastole; SNP, single-nucleotide polymorphisms; TBX5, T-box transcription factor 5; VSD, ventricular septal defect.

## Conclusions

Mice homozygous for Tbx5 R264K showed compensated dilated cardiomyopathy. Thus, TBX5 R264K may have a significant pathogenic role in some cardiomyopathy patients independently of T-box domain pathway.

## Introduction

Comprehensive genetic analysis of patients with genetic cardiomyopathies over the last decade have revealed significant genotype-phenotype associations. Most commonly, these genetic variants are single-nucleotide polymorphisms (SNPs), and may occur as familial or sporadic mutations. Identification of a genetic mutation in a patient with heart disease may provide useful information regarding prognosis, including the severity of the disease, morbidity or mortality and even therapeutic options. However, in many cases, the significance of the variant in pathogenesis is unknown and these are referred to as variants of unknown significance. Identification of a variant in multiple affected individuals may provide statistical evidence to support a causative role, but if a variant is rare it is difficult to establish a definitive role.

*TBX5* was identified as the gene responsible for Holt-Oram syndrome, a syndrome that is clinically characterized by radial ray deficiencies and cardiac defects, such as atrial septal defects (ASD) and ventricular septal defects (VSD). However, multiple variants in *TBX5* have been associated with Holt-Oram syndrome, with variations in cardiac morphology and severity of the defect [1–6]. Knockout or knockdown analysis of *Tbx5* in mice has revealed that the protein has a central role in cardiac development, probably through regulation of downstream genes, including *Anf*, *Gata4*, *Nkx2.5* and *Brg1* [1,7,8].

Although the effect of mutations in the T-box domain of TBX5 have been widely studied, the mechanism of mutations outside of the T-box domain have not and their role in disease not clearly established. With the development of genome editing technology, it is now much simpler to create SNP knock-in animal models.

Here, we report the identification of a missense TBX5 variant R264K that is over-represented in patients with left ventricular non compaction (LVNC) or dilated cardiomyopathy (DCM) compared with controls. The variant is located outside of T-box domain. Therefore, the aim of this study is to analyze whether this *TBX5* variant affects heart development and function using a knock-in mouse model.

## Materials and methods

### Collection of patient data

Basic clinical data were collected on patients at the time of enrollment. After the identification of genetic variants, detailed clinical data were retrospectively collected. The genetic testing of patients and use of the clinical data for the study were approved by the Ethics Committee of the University of Toyama and informed consent was obtained from all enrolled patients, or their parents.

**Patient genetic testing.** We designed a custom AmpliSeq panel of PCR primers (Life Technologies, Carlsbad, CA, USA) using Ion AmpliSeq designer software (www.ampliseq. com) to target all exons of 73 genes associated with cardiac disorders, including cardiomyopathies and channelopathies (S1 Table). This custom panel, which consisted of two separate PCR primer pools and produced a total of 1870 amplicons, was used to generate target amplicon libraries. Genomic DNA samples were PCR-amplified using the custom-designed panel and

Ion AmpliSeq Library Kit Plus (Life Technologies) according to the manufacturer's instructions. Samples were tagged using an Ion Xpress Barcode Adapters Kit (Life Technologies) and then pooled in equimolar concentrations. Emulsion PCR and Ion Sphere Particle (ISP) enrichment were performed using an Ion PGM Hi-Q View OT2 200 Kit (Life Technologies), according to the manufacturer's instructions. ISPs were loaded on an Ion 318 Chip v2 and sequenced with the PGM™ system using an Ion PGM Hi-Q View Sequencing Kit (Life Technologies). All variants derived from PGM sequencing were prioritized and then confirmed by Sanger sequencing to validate the next-generation sequencing (NGS) results. For Sanger sequencing, the nucleotide sequences of the amplified fragments were analyzed by direct sequencing in both directions using a BigDye Terminator v3.1 Cycle Sequencing Kit (Life Technologies) and an ABI 3130xl automated sequencer (Life Technologies).

**Data analysis of genetic testing.**   Torrent Suite software and Ion Reporter Software 5.1 (Life Technologies) were used to perform primary to tertiary analyses, including optimized signal processing, base calling, sequence alignment, and variant analysis.

**Pathogenic analysis *in silico*.**   SNP allelic frequencies were determined from the Human Genetic Variation Database of Kyoto University and the Genome Aggregation Database (gnomAD) (http://gnomad.broadinstitute.org), and multiple *in silico* algorithms were utilized to predict whether variants were likely deleterious and could contribute to disease pathogenesis.

## Generation of *Tbx5*$^{R264K/+}$ and *Tbx5*$^{R264K/R264K}$ mice

C57BL/6NJcl (CLEA Japan, Tokyo, Japan) strain mice were used in this study. Founder mice carrying a heterozygous R264K in *Tbx5* (*Tbx5*$^{R264K/+}$) were generated using CRISPR-Cas9 systems. The Slc:ICR (Japan SLC, Shizuoka, Japan) strain mouse was used as a surrogate mother, transplanted with an *in vitro* fertilized egg. Detailed methods are provided in the S1 File. Genotyping of offspring was performed by PCR using primers: 5'–AGGCTCAGCAAGGAGGTGAA–3' and 5'–CATGGCAGCGAGCAGTAAGG–3' followed by DNA sequencing. Homozygous mice (*TBX5*$^{R264K/R264K}$) were obtained by mating of founder *Tbx5*$^{R264K/+}$ mice. All animals were maintained under standard laboratory conditions (12–12 h/light–dark cycle with light on at 7:00 am; temperature, 22 ± 2°C; humidity, 50 ± 10%), and had free access to food and water, in the Laboratory Animal Resource Center of the University of Toyama. Mice were fed with a chow diet and housed in a barrier facility. All animal experiments were approved by the Animal Experiment Committee of the University of Toyama (Authorization No. A2017UH-2), and were carried out in accordance with the Guidelines for the Care and Use of Laboratory Animals of the University of Toyama.

## Analysis of cardiac function and histological change

**Echocardiography recordings.**   We used 10 week old *Tbx5*$^{R264K/R264K}$ mice as the young adult model and 6–9 month old *Tbx5*$^{R264K/+}$ and *Tbx5*$^{R264K/R264K}$ mice as the mature to middle age model, based on the definition in *The Mouse in Biomedical Research 2nd Edition* (2007) [9]. Age-matched wild-type mice were used as controls. Sedation and echocardiography were performed as previously described [10]. Sedation was conducted using 5% isoflurane with 1L/min airflow rate in the induction chamber followed by maintenance of sedation using 1–2% isoflurane, maintaining a steady heart rate of between 400–500 beats per minutes.

Echocardiography was performed using a SONIMAGE HS1 ultrasound system (KONICA MINOLTA, Inc.) and ultrawide waveband linear probe (L18-4) with mice in the supine position. The probe, with pre-warmed ultrasound gel, was gently placed on the left side parasternal thorax for the M-mode short axis view, including anterolateral and posteromedial papillary muscles, in order to measure systolic function and left ventricle diameter. In order to measure

diastolic function, trans-mitral inflow data were obtained by pulse-wave Doppler on an apical four chamber view. The following measurements were recorded: left ventricle diameter at end diastole and systole (LVDd and LVDs), left ventricular anterior and posterior wall thickness at end diastole (AWD and PWD), fractional shortening (FS) and trans-mitral early wave (E) and atrial wave (A).

These measurements were performed by at least two individuals, including pediatric cardiologists, and measurements were taken over an average of three heart beats. After echocardiography, the mice were immediately euthanized by intraperitoneal injection of sodium pentobarbital (100 μg/g body weight) and their hearts were harvested. Heart weight (HW) was measured immediately. Although body size or heart size varied between individual mice, HW was standardized by dividing by body weight (BW). Cardiac defects were detected during echocardiography or during dissection for histological sections.

**Isoproterenol stimulation to create myocardial injury.**   To examine cardiac tolerance in a stress environment, we used the isoproterenol-induced myocardial injury model as previously described [11–14]. Isoproterenol hydrochloride, a synthetic non-selective β-adrenergic agonist, was dissolved in physiological saline. Eight week old wild-type and $Tbx5^{R264K/R264K}$ mice were divided into two groups: isoproterenol (20 μg/g BW) group or isotonic saline group. Each group was injected subcutaneously once daily for 7 days. Seven days after the last injection, cardiac function was evaluated using echocardiography, as described above.

**Histological analysis.**   Harvested hearts were stored at -80˚C or fixed by immersion in 4% paraformaldehyde fixative. Fixed histological sections were stained with hematoxylin and eosin (HE) and Elastica van Gieson and Masson's trichrome (Elastica-Masson) stains. Since it is known that the progression of myocardial fibrosis has two patterns of reactive fibrosis and replacement fibrosis, we performed semi-quantitatively scoring of Elastica-Masson stained sections according to the previously reported method [15], where the fibrosis score was defined as 0 = negative, 1 = perivascular, 2 = perivascular with extension to interstitium, which was finely classified; a = focal and b = diffuse, and 3 = encircling of individual myocytes, which was classified; a = without perivascular fibrosis and b = with perivascular fibrosis. Similarly, replacement fibrosis was scored 0 = negative, 1 = minimal foci, 2 = occasional foci and small scars, and 3 = extensive scarring (S2 Table). This semi-quantitative method has shown a high correlation between computerized quantitative measurements in cardiac fibrosis [15].

**Electrocardiogram recordings.**   Electrocardiogram recordings were performed in 10 $Tbx5^{R264K/+}$ mouse and 10 wild-type mice at 4 weeks of age. Anesthesia was provided by intraperitoneal injection of 0.02 ml/ g BW of a mixture containing 2,2,2-tribromoethanol (0.0175 g/ml), 2-methyl-2-butanol (1% v/v) and physiological saline. Needle electrodes were inserted into skin, and the electrocardiogram was recorded for 5 minutes in leads  and  on spine position using the LabChart 7 ECG software module (AD Instruments, Australia). The average waveform of 4 consecutive heart beats was recorded to remove baseline noise, then the average of 10 times measurements was used for analysis. The measurements were heart rate, interval of PR, QRS, QT, and amplitude of P, Q, R, ST and T wave. Arrhythmias, such as atrioventricular block or sinus node dysfunction, were automatically and visually detected.

**Analysis of gene expression variation.**   *Microarray and real time RT-PCR analysis of RNA expression.*Total RNA was extracted from the left ventricular apex to the free wall of young adult $TBX5^{R264K/R264K}$ mice and wild-type littermates, using RNeasy Mini Kits (QIAGEN), according to the manufacturer's instructions. The quality and quantity of RNA was measured using an Eppendorf BioPhotometer D30 (Eppendorf) and the RNA was stored at -80˚C for future processing. For transcriptome analysis, Clariom™ D Arrays mouse (Thermo Fisher Scientific) was used, according to the manufacturer's instruction. RNA expression was analysed using the Transcriptome Analysis Console software (v.4.0.1.36) (Thermo Fisher Scientific) and

expressed as quantitative $\log_2$ metrics. Regarding the extracted genes, to explore the biological role of genes showing altered expression, functional enrichment analysis was performed using g:Profiler, a web server for functional enrichment analysis and conversions of gene lists [16]. For real time RT-PCR, cDNA synthesis was performed from 300 ng of RNA using Super Script Reverse Transcriptase (Invitrogen). Real time RT-PCR was performed using a THUNDER-BIRD® SYBR qPCR Mix (TOYOBO CO., LTD. Life Science Department, OSAKA, JAPAN) and all real time RT-PCR experiments were performed in duplicate. Expression levels were measured using a Thermal Cycler Dice Real Time System(Takara Bio) and compared by threshold cycle difference from target gene to *Gapdh* (ΔCt). Real time RT-PCR primers are shown in S3 Table.

*Western blotting.*Total protein from left ventricle tissue was prepared using PRO-PREP™ Protein Extraction Solution (Cell/Tissue) (iNtRON Biotechnology, Inc), according to the manufacturer's instruction. Protein extracts (10 μg) were separated by SDS-PAGE on 10% gel. Proteins were transferred to PVDF membranes, and the membranes were incubated with anti-skeletal muscle actin antibody (Mouse monoclonal LS-C413476 LifeSpan BioSciences, Inc) (1:5000) or anti-GAPDH antibody (Mouse monoclonal G8795 Sigma-Aldrich) (1:200) overnight at 4˚C. The membranes were then incubated with Goat Anti-Mouse (H+L)-HRP conjugate (Bio-Rad) (1:5000) for 1 hour at room temperature. Protein bands were detected using Luminata Forte Western HRP Substrate (Merck Millipore) and a ImageQuant LAS 4000 mini (GE Healthcare) system. Protein levels of ACTA1 were normalized to those of GAPDH.

## Analysis of transcriptional activity by luciferase assay

**Plasmid constructs.** A human TBX5 expression vector (IRAK049F12) was provided by the RIKEN BRC through the National Bio-Resource Project of the MEXT, Japan [17]. A series of *TBX5* variant plasmids containing single-base substitutions at different positions were constructed using the PrimeSTAR Mutagenesis Basal Kit (Takara, Shiga, Japan). For the human *NPPA* (*ANF*) promoter reporter construct, the promoter region of *NPPA* gene, -20 base to -758 base from the ATG start codon, was amplified by PCR and cloned into the pGL3-basic vector (Promega) using an In-Fusion HD Cloning Kit (Takara). All plasmids containing the various mutations were validated by direct DNA sequencing.

**Cell culture and DNA transfection.** Human embryonic kidney (HEK) 293T cells (CRL-11268, American Type Culture Collection, VA, USA) were grown in Dulbecco's modified Eagle's medium, supplemented with 10% fetal bovine serum and 0.5% penicillin-streptomycin, at 37˚C and 5% $CO_2$. Briefly, $2.0 \times 10^5$ HEK-293T cells were seeded into a 6-well plates. Twenty-four hours later, cells were transiently transfected with pCMV-SPORT6-*TBX5* (1.0 μg/well) or pCMV-SPORT6-*TBX5* mutant (R264K and P85T) (1.0 μg/well), and pGL3-vector (1.5 μg/well) or the pGL3-ANF promoter reporter (1.5 μg/well) vectors using TransIT-293 transfection reagent (Mirus, WI, USA). A previously reported *TBX5* P85T mutant was included in the transactivation study as a positive control [18]. A β-galactosidase control vector (the RSV LTR/β-galactosidase plasmid DNA) (50 ng) was used as an internal control. The total amount of transfected plasmids was adjusted to 2.55 μg/well by adding the pCMV-SPORT6 plasmid. At 48 h post-transfection, cells were harvested for luciferase and β-galactosidase assays. Cell lysis and luciferase assays were performed using the Luciferase Assay System (Promega, WI, USA). Light emission was measured in a SpectraMax i3 (Molecular Devices, CA, USA). The values were obtained in relative light units (RLU). Variations in transfection efficiency were normalized to the activities of luciferase expressed from the co-transfected β-galactosidase control vector. β-Galactosidase activities were measured using the β-

Galactosidase Enzyme Assay System (Promega). Activity of the pGL3 -ANF promoter vector was assigned an arbitrary value of 1.0.

## Statistical analysis

Student's *t*-test, or one- or two-way analysis of variance (ANOVA) with Tukey's multiple comparisons as a post-hoc analysis, were used to determine significance of quantitative data. Statistical significance was set at $P < 0.05$. Data are expressed as the mean ± standard error.

## Result

### Patient characteristics

A total of 233 idiopathic patients (192 LVNC and 41 DCM) have been screened using NGS technologies at our institution. Among them, 5 patients (4 LVNC and 1 DCM) had a heterozygous missense variant (c.791G>A, R264K) in *TBX5* (Table 1). The frequency of this variant in these patients (0.02) is significantly higher than in reported for Japanese individuals in the Human Genetic Variation Database (0.0075) and in gnomAD (0.0016). In addition, this variant was predicted to be likely pathogenic by multiple *in silico* algorithms (S4 and S5 Tables). None of these 5 patients had variants in any of the other genes previously associated with cardiomyopathy. LVNC was diagnosed in patients presenting with heart failure, heart murmur associated with a VSD or suspicion of anomalous cardiomyocyte maturation between the neonatal and infantile periods. The patient diagnosed with DCM presented with chest pain. Two patients presented with heart failure, one had severe heart failure and another complicated by an embolism that resulted in death at one month of age. One patient with LVNC had family history that included a paternal uncle who had died at 19 years old. The patient with the VSD had no other heart defects. Since cardiomyopathy due to *TBX5* variant is rare, we made heterozygous and homozygous knock-in mice to confirm the genotype-phenotype correlation between *TBX5* R264K variant and cardiomyopathies.

### *Tbx5*<sup>R264K/R264K</sup> mice developed compensated dilated cardiomyopathy

There was no significant difference in physique between *Tbx5*<sup>R264K/R264K</sup> mice and wild-type mice. Young adult *Tbx5*<sup>R264K/R264K</sup> mice displayed significantly reduced FS, AWD and PWD compared with wild-type littermates (Table 2A), as well as higher LVDd, LVDs and HW/BW ratios. No arrhythmias were detected during echocardiography and no cardiac defects, such as ASDs or VSDs, were detected during dissection or echocardiography. The changes in cardiac function were not related to gender. Furthermore, the isoproterenol administration test for assessment of myocardial tolerance resulted in no significant differences in FS between the two groups (Table 2B). From these data, it was concluded that the young adult *Tbx5*<sup>R264K/R264K</sup> mice developed dilated cardiomyopathy phenotype, but were within a compensable range.

As the homozygous mutant mice aged, their FS remained significantly reduced compared to wild-type mice but the differences HW/BW, LVDd, LVDs, AWD, and PWD were no longer significant (S6 Table).

Heterozygous mutant mice, *Tbx5*<sup>R264K/+</sup>, had no significant changes on echocardiography or in heart weight. Further, electrocardiographic examination of heterozygous mice revealed no differences in HR, interval of PR, QRS, QT, and amplitude of P, Q, R, ST, and T waves (S7 Table).

**Table 1. Characteristics of the patients carrying the heterozygous *TBX5* c.791G>A, R264K variant.**

| patient | sex | diagnosis | Age | gene variant in NGS associated with cardiomyopathy | opportunity | familial or sporadic | congenital heart disease |
|---|---|---|---|---|---|---|---|
| 1 | F | LVNC | neonatal | | screening | sporadic | VSD |
| 2 | M | LVNC | 3m | | heart failure | sporadic | none |
| 3 | F | LVNC | neonatal | | heart failure, embolus | familial | none |
| 4 | M | LVNC | neonatal | | fetal myocardial hypertrophy | sporadic | none |
| 5 | F | DCM | 11y | - | chest pain | sporadic | none |

DCM: dilated cardiomyopathy, LVNC: left ventricular non compaction, VSD: ventricular septal defect, -: no relative gene variant was detected

## Mild cardiac fibrosis in *Tbx5*$^{R264K/R264K}$ mice developing in an age-dependent manner

Young adult *Tbx5*$^{R264K/R264K}$ mice revealed slight fibrosis around vessels compared with wild-type on Elastica-Masson stain, with a reactive fibrosis score of 1. However, mature to middle aged *Tbx5*$^{R264K/R264K}$ mice had mild fibrosis from the endocardium to gaps between cardiomyocytes, and increased perivascular fibrosis with extension to the interstitium whereas mature to middle aged wild-type mice had only perivascular fibrosis. No other significant microscopic changes, such as hypertrophy of myocardial cells, difference in size or prolonged wavy appearance, were observed through histological analysis (Fig 1A–1H and Table 3).

## *Tbx5*$^{R264K/R264K}$ displayed significantly upregulated *Acta1*, specifically the isoform lacking exon 2

Microarray analysis of RNA isolated from the hearts of young adult *Tbx5*$^{R264K/R264K}$ mice revealed 36 genes (22 upregulated and 14 down regulated) differentially expressed (greater than 2.0 or less than -2.0 fold) by comparison with wild-type mice (Fig 2A and 2B and Table 4). After these were applied to g:Profiler analysis, noncoding genes were excluded, and as a result further analyses were limited to 8 coding genes, all of which were up-regulated genes in mutant mice (Table 4). Of the Biological Processes identified, there were two terms related to myocardium; response to muscle stretch and response to mechanical stimulus (S8 Table): this limited the relevant genes to *Acta1*, *Myot*, *Ankrd1* and *Ankrd23*. Downregulated genes were non-coding genes, small RNAs or micro RNAs, the significance of which are unknown at this time. Exon splicing variant analysis of *Acta1* showed that the significant variation derived from sample level signals was a 4.37-fold decrease in exon 2 in *Tbx5*$^{R264K/R264K}$ mice compared with wild-type mice (Fig 3).

By real time RT-PCR, the expression of 7 of the 8 genes was not significantly different (S1 Fig). However, the expression level of *Acta1* by real time RT-PCR was about 2 times higher in *Tbx5*$^{264K/R264K}$ mice than in the wild-type mice and ACTA1 protein expression was significantly higher by western blot analysis (Figs 4, 5A and 5B, and S2 Fig). No significant changes in the expression of *Tbx5* itself and *Nppa* as a load marker for the heart, were detected (S3 Fig). Since there were no significant changes other than Acta1, we also performed real time RT-PCR on other genes related to sarcomere or calcium ion kinetics. However, no differences in the following genes were observed; *myosin heavy chain 7 (Myh7)*, *myosin binding protein C3 (Mybpc3)*, *actin alpha cardiac muscle 1 (Actc1)*, *sarcolipin (Sln)* and *myosin light polypeptide 7 regulatory (Myl7)* (S4 Fig).

**Table 2.** **A.** Characteristics and echocardiographic data in young adult wild-type and *Tbx5*$^{R264K/R264K}$ mice. **B.** Characteristics and echocardiographic data of isoproterenol stimulation in young adult wild-type and *Tbx5*$^{R264K/R264K}$ mice.

**A**

| | Wild-type | *Tbx5*$^{R264K/R264K}$ |
|---|---|---|
| | (n = 5) | (n = 7) |
| male | 2 | 3 |
| BW (g) | 22.5±1.9 | 22.9±1.5 |
| HR (/min) | 413.2±12.7 | 431.2±34.2 |
| LVDd (mm) | 3.54±0.16 | 3.89±0.21* |
| LVDs (mm) | 2.14±0.23 | 2.56±0.17** |
| AWD (mm) | 0.68±0.12 | 0.56±0.05* |
| PWD (mm) | 0.82±0.07 | 0.66±0.1* |
| FS (%) | 39.6±5.4 | 34.2±1.7* |
| E (cm/s) | 51.9±7.0 | 61.2±14.1 |
| A (cm/s) | 31.2±6.3 | 34.1±5.0 |
| HW (mg) | 115.4±10.2 | 132.9±8.9* |
| HW/BW | 5.12±0.08 | 5.80±0.32** |

**B**

| | Wild-type | | *Tbx5*$^{R264K/R264K}$ | |
|---|---|---|---|---|
| | Vehicle (n = 4) | Isoproterenol (n = 4) | Vehicle (n = 4) | Isoproterenol (n = 4) |
| male | 2 | 2 | 2 | 2 |
| BW (g) | 24.5±1.0 | 24.7±0.9 | 22.8±1.5 | 23.4±0.8 |
| HR (/min) | 430.7±9.2 | 429.6±15.1 | 443.7±14.4 | 444.7±14.1 |
| LVDd (mm) | 3.63±0.22 | 3.83±0.11 | 4.10±0.25 | 4.18±0.09§ |
| LVDs (mm) | 2.33±0.17 | 2.66±0.11 | 2.90±0.2 | 3.18±0.09§ |
| AWD (mm) | 0.85±0.06 | 0.90±0.06 | 0.63±0.05† | 0.73±0.03§ |
| PWD (mm) | 0.95±0.09 | 0.88±0.05 | 0.58±0.05†† | 0.7±0.04§ |
| FS (%) | 36.0±1.1 | 30.4±1.7* | 29.4±0.8†† | 24.0±0.5‡§§ |
| E (cm/s) | 55.8±2.7 | 47.5±1.4* | 57.5±3.7 | 48.8±2.5 |
| A (cm/s) | 35.8±5.8 | 27.5±3.9 | 33.1±3.5 | 34.9±3.6 |
| HW (mg) | 117.0±5.5 | 119.0±4.3 | 116.3±5.9 | 125.5±3.9 |
| HW/BW | 4.45±0.11 | 4.82±0.18 | 5.11±0.10†† | 5.37±0.04§ |

BW: body weight, HR: heart rate, LVDd: left ventricle diameter at end diastole, LVDs: left ventricle diameter at end systole, AWD: left ventricular anterior wall thickness at end diastole, PWD: left ventricular posterior wall thickness at end diastole, FS: fractional shortening, E: trans-mitral early wave, A: trans-mitral atrial wave, HW: heart weight, *P* value *<0.05, versus wild-type vehicle,

†<0.05,

††<0.01, versus wild-type vehicle,

‡<0.05 versus young adult *TBX5*$^{R264K/R264K}$ vehicle, and.

§<0.05,

§§<0.01, versus wild-type isoproterenol. *P* value.

*<0.05,

**<0.01.

## Transactivation of the ANF promoter was unaffected by the TBX5 variant

There were no significant changes in activation of the ANF promoter by the R264K mutation compared to wild-type TBX5: the positive control P85T mutant showed significantly reduced transcriptional activity (Fig 6).

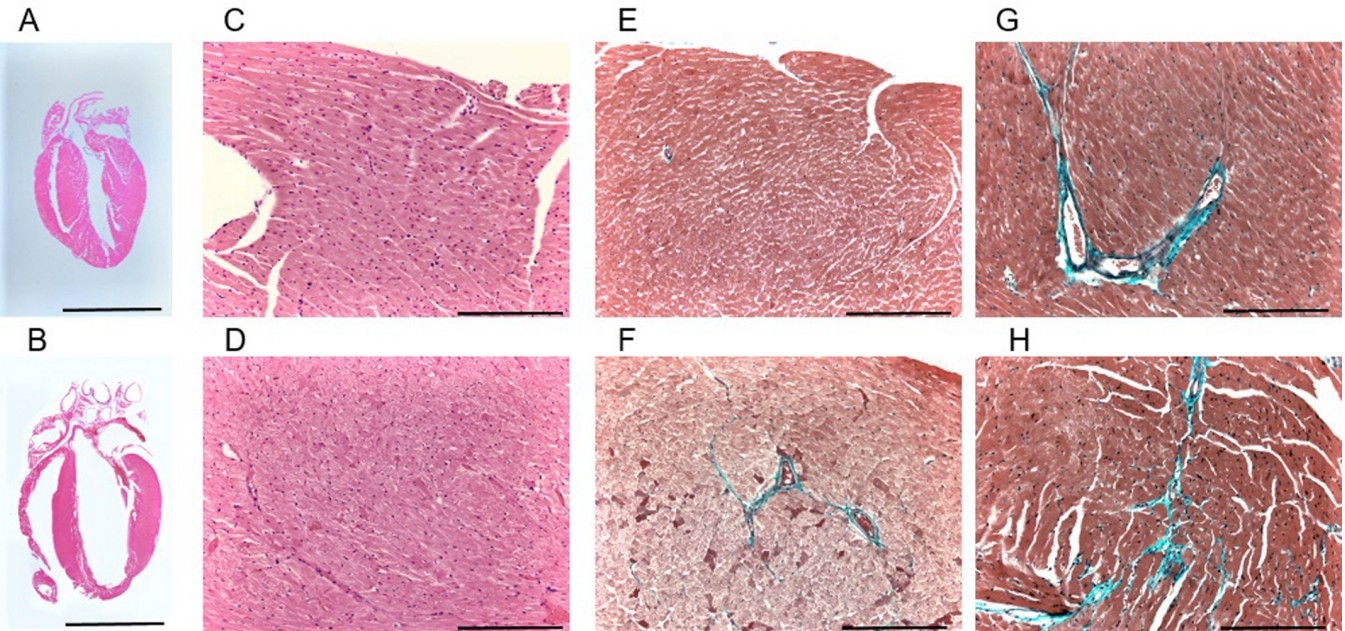

**Fig 1. Histological findings in wild-type (upper panels) and *TBX5*[R264K/R264K] mice (lower panels).** A and B. Longitudinal sections through the atria and ventricles with hematoxylin-eosin staining. Scale bars: 2 mm. C and D. Hematoxylin-eosin staining of high magnification sections of left ventricle myocardium. Scale bars: 100 μm. E and F. Elastica-Masson staining of sections from the left ventricular apex to free wall of hearts from young adult mouse. Slight fibrosis (blue signals) is seen around the vessels in young adult *TBX5*[R264K/R264K] mouse. Scale bars: 100 μm. G and H. Elastica-Masson staining of sections from the left ventricular apex to free wall of hearts from mature to middle age mice. Mild fibrosis (blue signals) from the endocardium to gaps between cardiomyocytes, and increased perivascular fibrosis with extension to the interstitium are shown in *TBX5*[R264K/R264K] compared with wild-type. Scale bars: 100 μm.

## Discussion

In this study, we found that homozygous *Tbx5*[R264K/R264K] mice exhibited cardiac dysfunction mimicking dilated cardiomyopathy. In particular, young adult *Tbx5*[R264K/R264K] mice had decreased FS, dilatation of the left ventricle, left ventricular wall thinning, and heart weight increases. We also showed that young adult *Tbx5*[R264K/R264K] mice had significant up-regulation of expression of *Acta1*.

TBX5 has been shown to be an essential transcription factor for cardiac development. *Tbx5* heterozygous knockout mice develop cardiac defects, such as ASDs or VSDs, due to heart developmental patterning defects [19,20]. These result from significantly decreased expression of *Cx40* and mis-expression of *ANF* into the right ventricle and the intraventricular groove rather than limited to the left ventricle, as seen in wild-type mice. *Tbx5* homozygous hypomorphs develop hypoplastic left ventricles, resulting in embryonic lethality at E10.5 [19,20]. These structural defects often cause conduction diseases, such as arrhythmia and atrioventricular block [3,21,22]. In addition, *TBX5* is considered to play a role in diastolic function, through $Ca^{2+}$ handling modulation [23–25]. *TBX5* variants have been identified in patients

**Table 3. Semi-quantitative evaluation of myocardial fibrosis based on S2 Table in wild-type and *Tbx5*[R264K/R264K] mice.**

|  | Young adult | | Mature to middle age | |
| --- | --- | --- | --- | --- |
|  | **Wild-type** | ***Tbx5*[R264K/R264K]** | **Wild-type** | ***Tbx5*[R264K/R264K]** |
| Reactive fibrosis | 0 | 1 | 1 | 2a |
| Replacement fibrosis | 0 | 0 | 0 | 1 |

A

## Normalized probe intensity

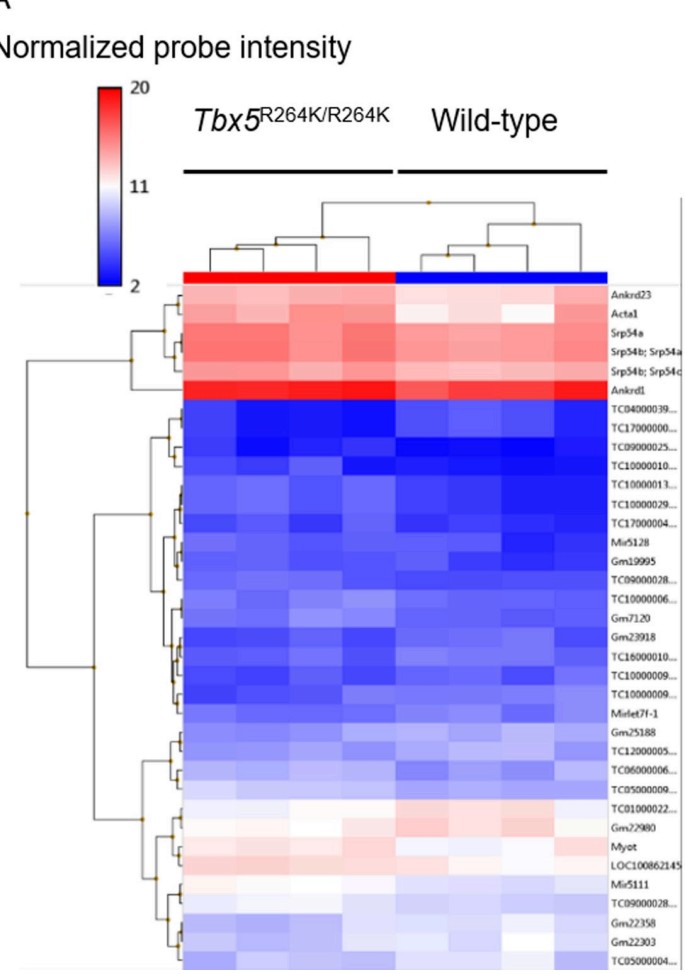

B

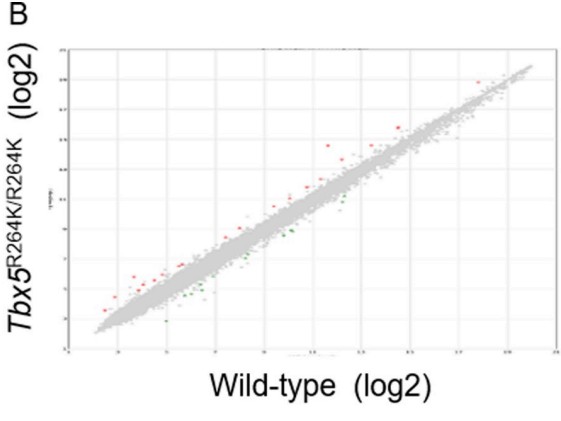

**Fig 2. Microarray analysis of RNAs isolated from the left ventricle myocardium of young adult *Tbx5*^R264K/R264K mice compared with wild-type.** A Clustering map shows 36 RNAs were significantly (*P* value <0.05) changed with greater than 2.0 or less than -2.0 fold expression changes. B Scatter plot shows those 36 RNAs indicated as red (increased) and blue (decreased) points.

with congenital cardiac defects, with most located in the T-box domain (S5 Fig) [26–29]. However, there are only few reports of the identification of *TBX5* variants in patients with LVNC or DCM unlike congenital heart defects [2, 5,30]. There is not a significant relationship

**Table 4. Significant 8 mRNAs detected with microarray analysis, all of which were increased in young adult *Tbx5*^R264K/R264K mice, remaining after excluding non-coding genes from 36 RNAs.**

| Gene Symbol | Fold Change |
|---|---|
| *Gm7120* | 2.05 |
| *Myot* | 2.08 |
| *Ankrd1* | 2.12 |
| *Srp54b; Srp54c* | 2.32 |
| *Srp54a* | 2.44 |
| *Srp54b; Srp54a* | 2.47 |
| *Ankrd23* | 2.82 |
| *Acta1* | 7.92 |

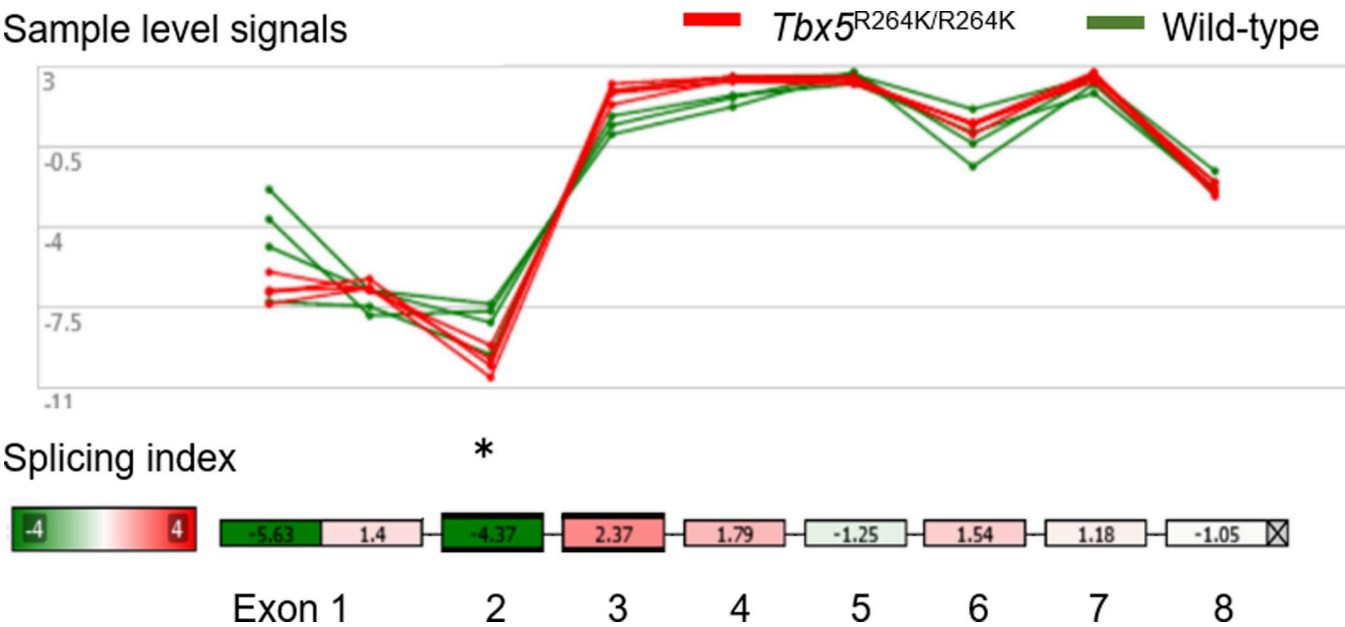

**Fig 3. Exon splicing variant analysis in *Acta1* by Transcriptome Analysis Console software.** Transcriptome Analysis Console software shows the sample level signals and schema of *Acta1* for each exon from 1 to 8 of each 4 wild-type mice (green) and *Tbx5*^R264K/R264K^ mice (red) in the young adult group. Of these, the only significant variation derived from sample level signals was a 4.37-fold decrease in exon 2 in the *Tbx5*^R264K/R264K^ mice group. *P* value * < 0.05, versus wild-type.

between the cardiac phenotype and whether the variant is in the T-box or non T-box domain. A recent study revealed that a non T-box domain (amino acids 255–264) interacts with the nucleosome remodeling and deacetylase (NuRD) repressor complex (S5 Fig). This NuRD interaction pathway was considered to repress incompatible gene programs involved in cardiac development and, as a result, cardiac defects are inhibited [28].

In this context, we describe the analysis of a heterozygous *TBX5* R264K variant that is over-represented in cardiomyopathy patients. This variant lies within the *TBX5*-NuRD interaction domain and is predicted to be likely pathogenic by multiple *in silico* algorithms. Therefore, we aimed to investigate whether this variant affects in cardiac development and/or cardiac function.

We found that *Tbx5*^R264K/R264K^ mice develop DCM phenotype, without a major heart structural disorder. DCM is a chronic disease presenting left ventricular dilatation and systolic dysfunction, except for secondary reasons such as hypertension, valvular disease and coronary artery disease [31]. LVNC is defined by the feature of prominent trabeculae, intratrabecular recesses, and two distinct myocardial wall layers: a thin compacted epicardial layer and a thickened endocardial non-compacted layer [31]. In most patients with LVNC heart failure, the clinical characteristics are often similar to DCM in terms of left ventricular dilatation and systolic dysfunction. The findings in *Tbx5*^R264K/R264K^ mice were characteristic of compensable DCM without heart structural defects or arrhythmias.

As the mice aged, the deposition of fibrosis increased in the endocardium and interstitium. The reactive fibrosis occurs with angiogenesis against the load on cardiomyocytes, and the replacement fibrosis occurs to fill the gap of cardiomyocytes when the cells fall into ischemia or inflammation leading to necrosis [32]. These fibrosis events were observed in mature to middle age *Tbx5*^R264K/R264K^ mice. The slow progression of these histological changes is dependent on time with increasing contractile dysfunction, according to a mouse model of severe DCM [33]. Thus, the lack of changes, except for systolic dysfunction on echocardiography, in

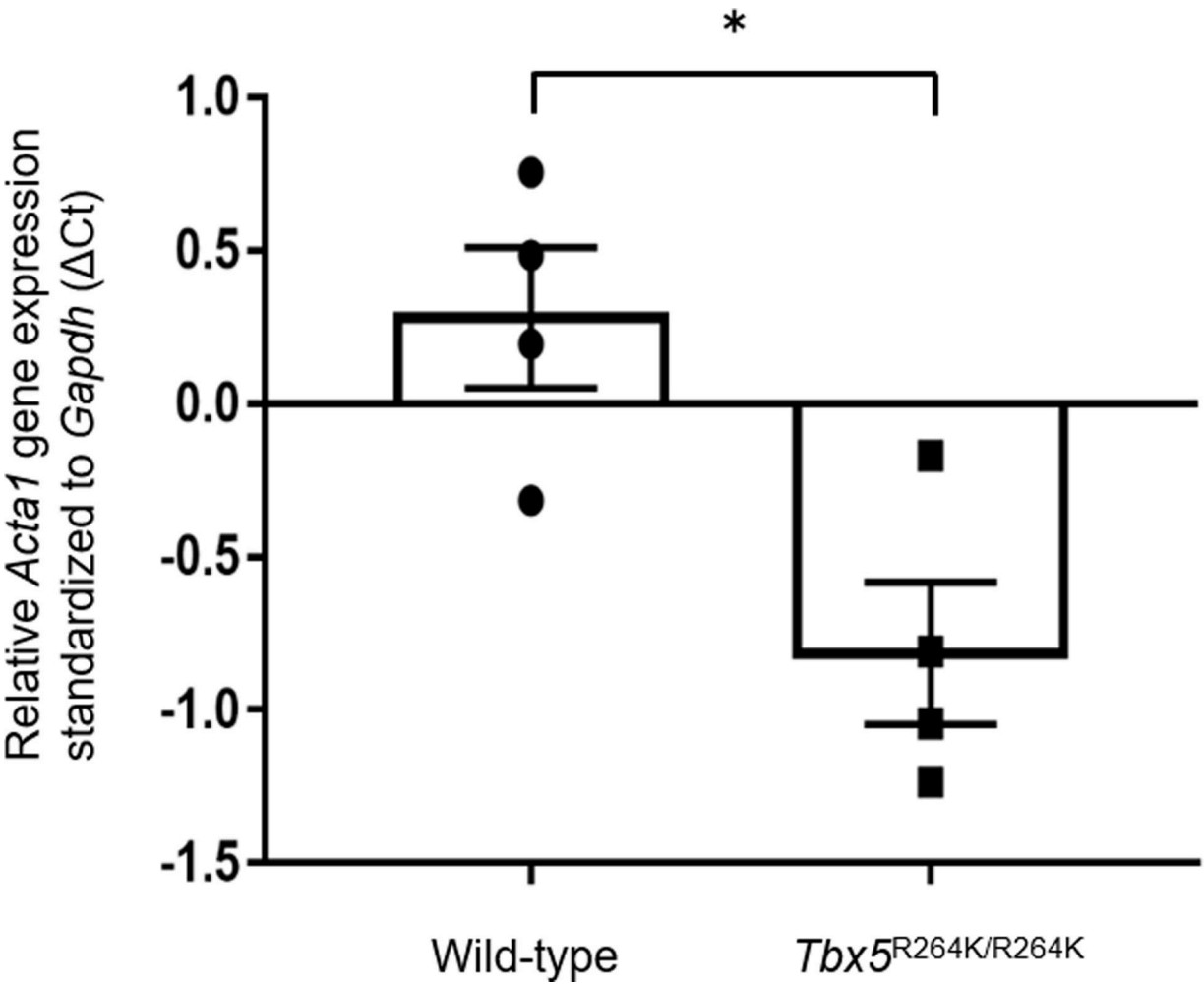

**Fig 4. Relative gene expression of *Acta1* in hearts of young adult *Tbx5*[R264K/R264K] and wild-type mice by real time RT-PCR analysis.** In the young adult *Tbx5*[R264K/R264K] mice group, the expression of *Acta1* mRNA was approximately 2 times higher (lower ΔCt value) than in the wild-type mice group by real time RT-PCR. The number of individuals is four in each group. Average value of ΔCt (mean ± SEM) in each group is plotted. *P* value *<0.05, versus wild-type.

mature to middle aged *Tbx5*[R264K/R264K] mice was due to myocardial remodeling that occurred slowly as the mice aged.

Microarray analysis showed significant changes in the expression of 36 genes in the hearts of young adult *Tbx5*[R264K/R264K] mice compared with wild-type mice, and functional enrichment analysis based on these genes showed specific terms related to myocardium. However, there were no significant changes in expression detected by RT-PCR except for *Acta1*. Although the relationship between real time RT-PCR results and functional enrichment analysis could not be determined, it is presumed that the number of candidate genes was too small to investigate biological process. Moreover, the function of noncoding RNAs has not been fully elucidated. However, as there is a study describing noncoding RNA profiles in DCM patients [34], the noncoding RNAs identified in our study may also have important implications for cardiomyopathy pathogenesis.

*ACTA1* is regulated by *TBX5* during heart development, although it is expressed more highly in skeletal muscle than heart muscle [35]. *ACTA1* is the causal gene of nemaline

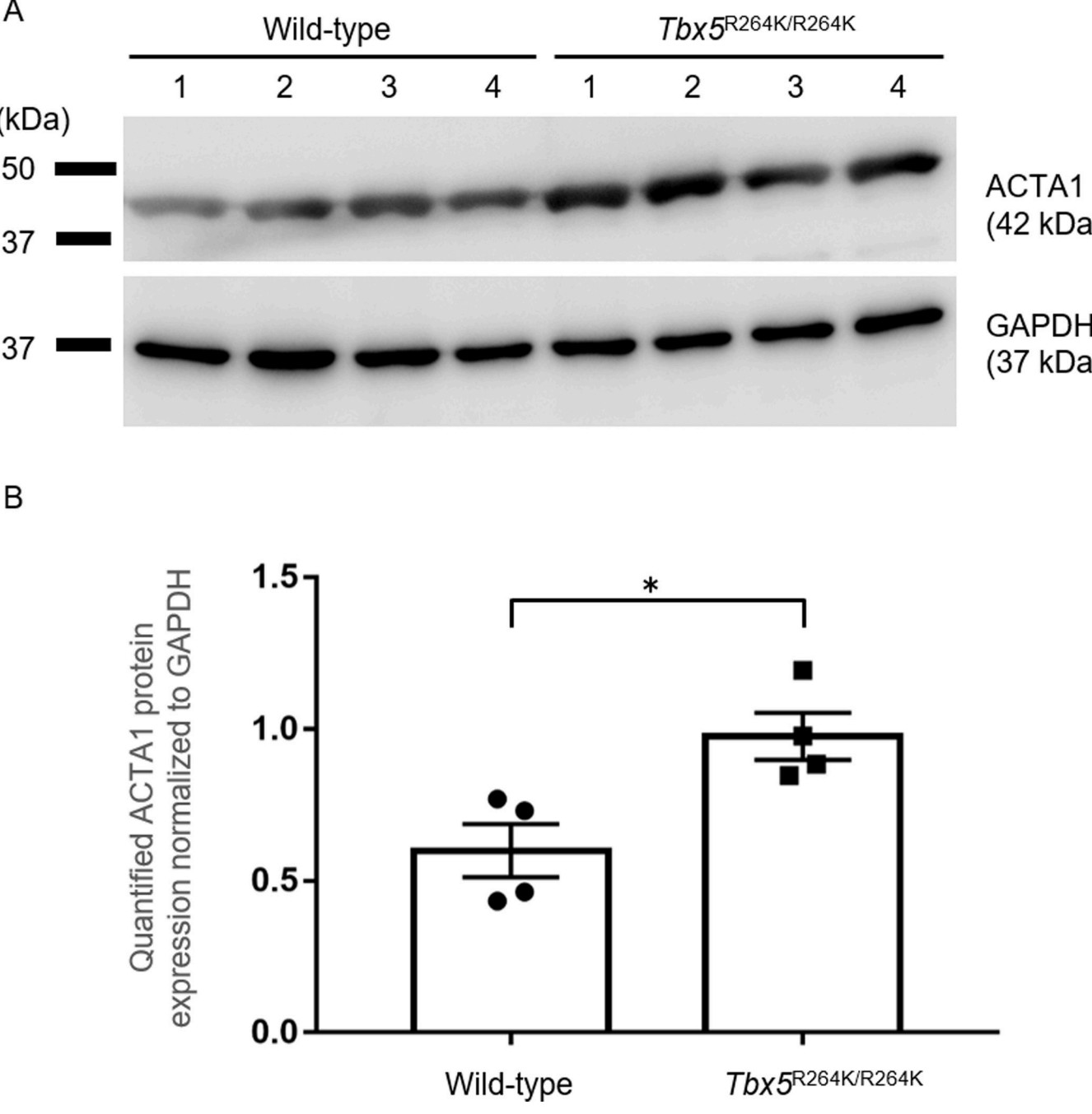

**Fig 5. Comparison of ACTA1 protein expression in hearts of young adult *Tbx5*^R264K/R264K^ and wild-type mice.** A Western blotting shows expression of ACTA1 protein (upper) in young adult *Tbx5*^R264K/R264K^ and wild-type mice. GAPDH (lower) is shown as loading controls. The number of samples is four respectively. B In the young adult *Tbx5*^R264K/R264K^ mice, the amount of ACTA1, normalized to GAPDH was about 1.6 times higher than that of in the wild-type mice. Quantified ACTA1 value normalized to GAPDH (mean ± SEM) is plotted. *P* value *<0.05, versus wild-type.

myopathy, characterized by an abnormal distribution of muscle filaments resulting in myopathy [36,37]. There have been reports of DCM patients with *ACTA1* mutations [38]. Elevated *ACTA1* expression has been reported in human heart failure samples and in a model mouse of DCM [39]. It has been concluded that increasing ACTA1 in heart failure is to compensate for

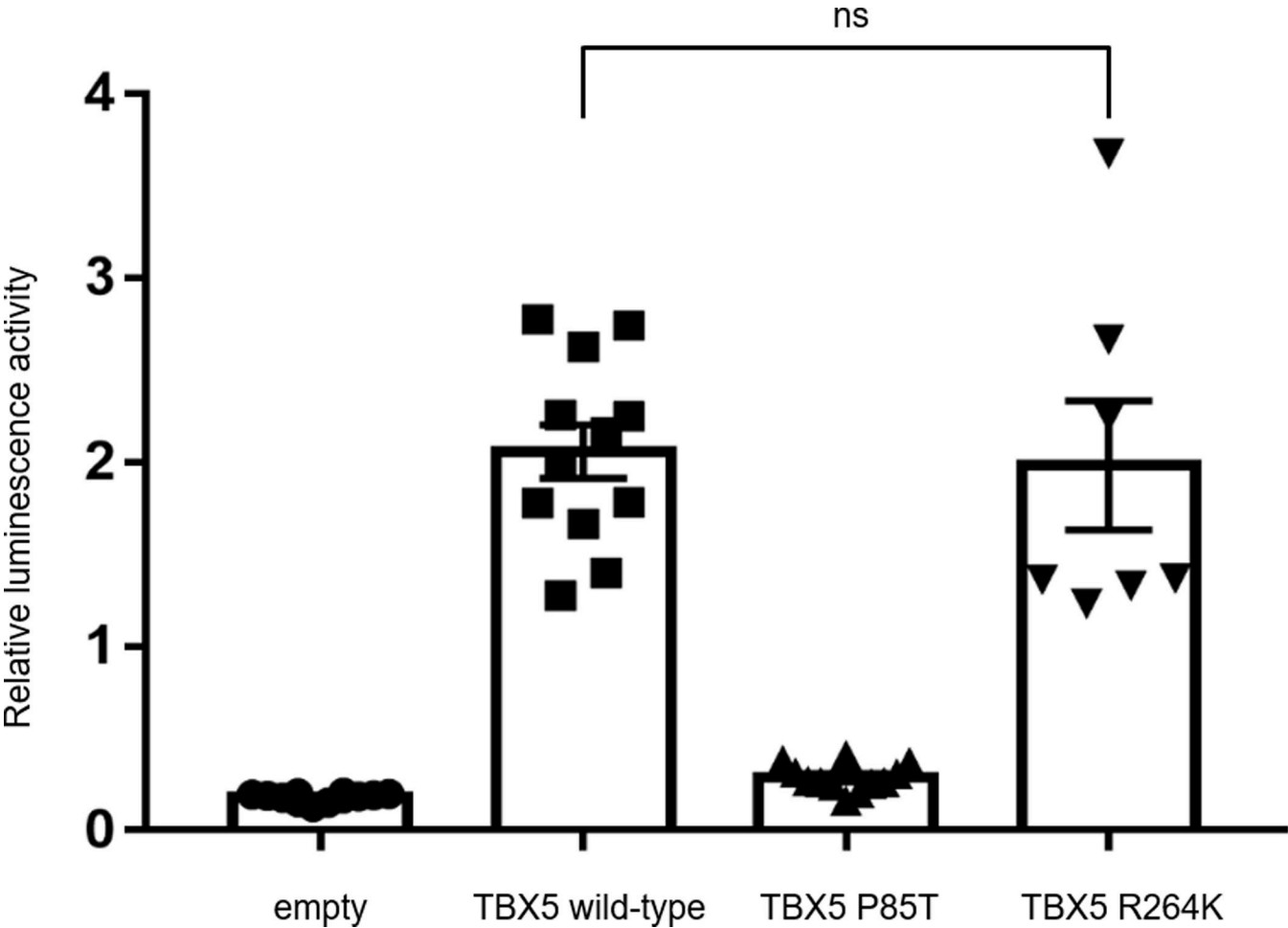

**Fig 6. Assays of TBX5 transcriptional activity.** *TBX5* activates transcription from the ANF promoter. HEK293T cells were cotransfected with the ANF reporter plasmid and plasmids overexpressing *TBX5*, *TBX5*P85T, *TBX5*R264K. The *TBX5*R264K mutant activated the ANF promoter in a manner similar to the wild-type *TBX5*. Relative luminescence activity (mean ± SEM) is plotted.

contraction failure, irrespective of cause [39]. Transcriptome Analysis Console software identified a splicing variant of *Acta1*, with significantly increased splicing of exon 2. The function of exon 2 of *Acta1* is unclear, though the exon encodes part of the 5' untranslated region, and, in general, untranslated regions are speculated to regulate protein translation, translation activity or mRNA expression [40–42]. The transcription factor Tbx5 R264K was functionally and quantitatively preserved showing *ANF* expression via T-box domain pathway. However, it has been previously reported that the non-T-box domain pathway associated with amino acids 255–264 has a role in DCM pathogenesis in mice [28].

Since we were unable to identify the underlying mechanism of DCM in *Tbx5*[R264K/R264K] mice by gene expression analysis, we investigated the effects of isoproterenol stimulation on phenotype. Excessive β-adrenergic receptor stimulation activates the mitogen-activated protein kinase cascade, causing remodeling of myocardial hypertrophy resulting in systolic or diastolic dysfunction. The role of specific gene mutations in diastolic dysfunction induced by isoproterenol stimulation has been reported [43]. If isoproterenol stimulation caused a significant cardiac dysfunction in the *Tbx5*[R264K/R264K] mice, it is likely that the *Tbx5*[R264K/R264K] caused fragility in cardiomyocytes associated with mutation-specific changes in gene

expression. However, no phenotypic differences were observed on isoproterenol stimulation, so we were unable to investigate possible changes in gene expression.

The patients with variants in the *TBX5* gene showed severe cardiomyopathy phenotypes despite heterozygosity, while the mice with homozygous variant barely presented with compensatory chronic, decreasing cardiac contractility. Therefore, the cause of cardiomyopathy in these patients is not likely to be *TBX5* R264K alone and it is likely that other unknown genes or environmental factors might have triggered those patients' diseases [44]. In other words, this variant could act as modifier of the phenotype.

There is a limitation to this study. We have not confirmed that the non-Tbox domain pathway: NuRD interaction pathway is actually impaired. Therefore, the genetic pathway that led to DCM in *Tbx5*[R264K/R264K] mice has not been clarified.

## Conclusion

We developed knock-in mice expressing the R264K Tbx5 variant that is frequently detected in cardiomyopathy patients. The homozygous variant of R264K in Tbx5 leads to compensatory DCM, without major heart structural disorders in mice, and ACTA1 expression was increased in cardiomyocytes in association with decreased contractility. Since the influence of other genes and environmental factors cannot be ignored, the TBX5 R264K variant in the non-Tbox domain involving NuRD interaction pathway may play a role in the pathogenesis of human cardiomyopathy.

## Supporting information

**S1 Fig. Relative gene expression of seven upregulated coding genes in microarray analysis of young adult *Tbx5*[R264K/R264K] and wild-type mice by real time RT-PCR analysis.**
(TIF)

**S2 Fig. Raw image data of western blotting of ACTA1 and GAPDH in young adult *Tbx5*[R264K/R264K] and wild-type mice shown in Fig 5.**
(TIF)

**S3 Fig. Relative gene expression of *Tbx5* and *Nppa* in young adult *Tbx5*[R264K/R264K] and wild-type mice by real time RT-PCR analysis.**
(TIF)

**S4 Fig. Relative gene expression of genes related to sarcomere or calcium ion kinetics in young adult *Tbx5*[R264K/R264K] and wild-type mice by real time RT-PCR analysis.**
(TIF)

**S5 Fig. Schematic representation of TBX5 and the location of domains.** The numbers below indicate the positions of previously reported amino acid substitutions associated with congenital heart disease (35 variants in all). The variants described in this study, R264K, is located within the NuRD Interaction Domain.
(TIF)

**S1 File. Detailed methods.**
(PDF)

**S1 Table. List of the genes associated with inherited cardiac disease that were analyzed by NGS.**
(PDF)

**S2 Table. Grading scale for reactive fibrosis and replacement fibrosis.**
(PDF)

**S3 Table. PCR primer sequences (5′ to 3′) used to real time RT-PCR.**
(PDF)

**S4 Table. Variant identified in the patients.**
(PDF)

**S5 Table. *In silico* predictive algorithms used in the study.**
(PDF)

**S6 Table. Characteristics and echocardiographic data in wild-type and *Tbx5*$^{R264K/R264K}$ mice in mature to middle age. *P* value $^*<0.05$.**
(PDF)

**S7 Table. Electrocardiogram data comparing *Tbx5*$^{R264K/+}$ and wild type mice.**
(PDF)

**S8 Table. Terms about biological processes inferred by g:Profiler.**
(PDF)

## Acknowledgments

The authors would like to thank Dr. Hideki Origasa, division of Biostatistics and Clinical Epidemiology, University of Toyama, Toyama, Japan for advice of statistical methods. The authors also would like to thank Mr.Hitoshi Moriuchi, technical assistant in department of pediatrics, University of Toyama for preparation and proxy purchase of laboratory equipment. We thank Dr. Yoshihide Hayashizaki at RIKEN and Dr. Sumio Sugano at University of Tokyo for providing us the human TBX5 expression vector (IRAK049F12). The authors are grateful to Dr.Neil E Bowles for assistance with editing.

## Author Contributions

**Conceptualization:** Nariaki Miyao, Hisashi Mori, Fukiko Ichida, Keiichi Hirono.

**Data curation:** Nariaki Miyao, Hironori Izumi, Keiichi Hirono.

**Formal analysis:** Nariaki Miyao, Hironori Izumi, Keiichi Hirono.

**Funding acquisition:** Fukiko Ichida, Keiichi Hirono.

**Investigation:** Nariaki Miyao, Yukiko Hata, Hironori Izumi, Ryo Nagaoka, Yuko Oku, Ichiro Takasaki, Taisuke Ishikawa, Shinya Takarada, Mako Okabe, Hideyuki Nakaoka, Keijiro Ibuki, Sayaka Ozawa, Hideyuki Hasegawa, Naomasa Makita, Naoki Nishida, Keiichi Hirono.

**Methodology:** Tomoyuki Yoshida, Hisashi Mori, Fukiko Ichida, Keiichi Hirono.

**Project administration:** Hisashi Mori, Fukiko Ichida.

**Resources:** Nariaki Miyao, Yukiko Hata, Ichiro Takasaki, Hideyuki Hasegawa, Naoki Nishida, Hisashi Mori, Fukiko Ichida, Keiichi Hirono.

**Software:** Nariaki Miyao, Yukiko Hata, Taisuke Ishikawa, Hideyuki Hasegawa, Naoki Nishida, Hisashi Mori, Fukiko Ichida, Keiichi Hirono.

**Supervision:** Hisashi Mori, Fukiko Ichida, Keiichi Hirono.

**Validation:** Nariaki Miyao, Yukiko Hata, Hironori Izumi, Ryo Nagaoka, Ichiro Takasaki, Hideyuki Hasegawa, Naomasa Makita.

**Visualization:** Nariaki Miyao, Yukiko Hata, Hironori Izumi, Keiichi Hirono.

**Writing – original draft:** Nariaki Miyao, Yukiko Hata, Hironori Izumi, Keiichi Hirono.

**Writing – review & editing:** Nariaki Miyao, Yukiko Hata, Hironori Izumi, Tomoyuki Yoshida, Hisashi Mori, Fukiko Ichida, Keiichi Hirono.

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
