## [Decision Letter · Decision Letter 0]

14 Jan 2020

PONE-D-19-34630

TBX5 R264K induces chronic heart failure not via T-box pathway

PLOS ONE

Dear Dr. Hirono,

Thank you for submitting your manuscript to PLOS ONE. After careful consideration, we feel that it has merit but does not fully meet PLOS ONE’s publication criteria as it currently stands. Therefore, we invite you to submit a revised version of the manuscript that addresses the points raised during the review process.

All issues raised by expert reviewers are required.

We would appreciate receiving your revised manuscript by Feb 28 2020 11:59PM. To enhance the reproducibility of your results, we recommend that if applicable you deposit your laboratory protocols in protocols.io, where a protocol can be assigned its own identifier (DOI) such that it can be cited independently in the future. For instructions see: http://journals.plos.org/plosone/s/submission-guidelines#loc-laboratory-protocols

We look forward to receiving your revised manuscript.

Kind regards,

Vincenzo Lionetti, M.D., PhD

Academic Editor

PLOS ONE

Journal Requirements:

2. Please modify the title to ensure that it is meeting PLOS’ guidelines (https://journals.plos.org/plosone/s/submission-guidelines#loc-title). In particular, the title should be "specific, descriptive, concise, and comprehensible to readers outside the field" and in this case it is not informative and specific about your study's scope and methodology.  When amending the title please amend this both on the online submission form (via Edit Submission) and in the manuscript so that they are identical.

https://academic.oup.com/eurheartj/article/29/2/270/433902

The text that needs to be addressed involves the Discussion.

In your revision ensure you cite all your sources (including your own works), and quote or rephrase any duplicated text outside the methods section. Further consideration is dependent on these concerns being addressed.

Reviewers' comments:

Reviewer's Responses to Questions

**Comments to the Author**

1. Is the manuscript technically sound, and do the data support the conclusions?

Reviewer #1: Yes

Reviewer #2: Yes

2. Has the statistical analysis been performed appropriately and rigorously? 

Reviewer #1: Yes

Reviewer #2: Yes

3. Have the authors made all data underlying the findings in their manuscript fully available?

Reviewer #1: No

Reviewer #2: No

4. Is the manuscript presented in an intelligible fashion and written in standard English?

Reviewer #1: Yes

Reviewer #2: Yes

5. Review Comments to the Author

Reviewer #1: Minor Comments

1. Authors are suggested to change the title from “TBX5 R264K induces chronic heart failure not via T-box pathway” to “TBX5 R264K induces chronic cardiomyopathy independently of T-box pathway”, so that it is more appealing to the reader.

2. Please avoid the use of the term heart failure in your in vivo experiments, as a NYHA scoring should be performed to claim it. Please change the term to cardiomyopathy, when referring to the murine models.

3. Table 2B: Please correct Ds to LVDs.

4. Figure 5, considering the Western Blot analysis of ACTA1 is not self explanatory. Proper labelling of the WB images is missing, as well as the molecular weight of the identified targets and the labelling of the samples. The authors are suggested to change it accordingly.

5. The resolution of the figures is poor, which creates an inconvenience for the reader. Authors are suggested to provide a higher image resolution to the given figures. Moreover, concerning the column plots, authors are suggested to use another software, rather than Microsoft Office Excel, in order to export more appealing plots and figures.

Major Comments

1. Authors state to have used transgenic mice of 6-9 months old as an aging model. However, according the literature, mice of that age correspond to human adults around 40 years of age. In order to consider a mouse aged, the minimum age of 18 months should be reached. Authors are advised either to correct the statement, or to present mortality data justifying the selection of the specific age for their murine model.

2. Hematoxylin-Eosin staining seems to be poorly performed as nuclei (blue) are not visible in the histology images. Please refine your staining, by improving your H&E protocol and by washing your specimens with PBS for the removal of blood (as visible in Fig. 1A lower panel). Moreover, considering that authors are commenting on intracardiac fibrosis in their transgenic mice, maybe another more specific staining should be applied such as Sirius Red or Masson Trichrome. By the use of these staining techniques the visualization of collagen deposition will be more straight-forward.

3. Column graphs are not presented in a sufficient manner. Individual values, showing the distribution and the number of the biological replicates should be present in the graphs (i.e. scatter plotted column graphs) and statistical analysis (i.e. asterisks) should be presented, deducing the statistically significantly differences between the groups.

4. The use of isoproterenol is not sufficiently justified. Transgenic mice, already present cardiac contractility deficits (e.g. decreased FS%) in the absence of the β-agonist. Therefore, the use of isoproterenol should be better discussed in the Discussion section, considering its transcriptional value in the experiments.

5. Authors shown that the given SNP in the TBX5 leads to a downregulation of ACTA-1 in the myocardium. Concerning that ACTA-1 is a highly important protein in the cellular cytoskeleton, authors are suggested to perform electron microscopical evaluation of the cardiomyocytes’ structure in the transgenic mice. Therefore, a more detailed representation of the effect of TBX5 SNP on cardiomyocytes’ architecture will be available.

6. If applicable, it would be interesting if the authors could confirm the results from their transgenic murine model in human cardiac biopsies, following simple RT-PCR or Western Blot experiments. Thus, the translational value of the current manuscript will be increased.

Reviewer #2: The manuscript describes a new variant of the TBX5 gene located outside the T-box regulatory domain. The variant is over-represented in cardiomyopathy patients and leads to compensated dilated cardiomyopathy in a knock-in animal model. The study is well conducted and organized. Although the new SNP does not seem to be sufficient to induce de-compensated heart disease by itself, the results indicate that it may work as an important phenotype modifier during the evolution of adverse cardiac remodeling in patients. However, the following points must be addressed.

Major points

-The authors focused part of the study on the contribution of differentially expressed mRNAs in the observed phenotypes, have the authors performed a functional enrichment analysis of the differentially expressed transcripts to explore putative underlying biological processes?

-The majority of the differentially expressed transcripts are still uncharacterized non coding RNAs. To foster future investigation, it should be interesting if the authors could briefly discuss their putative role in the observed phenotype.

-According to the results shown on page 13 lines 362-369, the differential expression of Myot between wt and mutant mice was not confirmed by PCR. However the authors seem to disregard this finding when discussing the putative role of Myot in disease evolution. Please clarify.

-Why only 2 out of 8 differentially expressed mRNAs were validated by rt-PCR? To confirm the microarray data all the 8 differentially expressed mRNAs must be validated by real time PCR.

-In some instances the authors report the findings as data not shown (see results lines 330, 334, 368). Although these results refer to not statistically significant differences, all the data must be fully available for the reader. Please amend.

-The authors observed a milder phenotype in the aged mutants with respect to the young-adult ones. How do they explain this finding?

- In the light of the limitations acknowledged by the authors themselves (see discussion 457-462), the title should be rephrased given more emphasis to the role of the mutation as a phenotype modifier rather that as an indipendent pathogenic mechanism.

-The quantification of the histological results shown in fig 1 is lacking and must be added.

-The quantification of the WB data shown in figure 5 is lacking and must be added.

-The plot shown in figure 6 is reported without indication of the statistical analysis. Although evident, the statistically significant differences must be indicated.

Minor points

-Abstract, line 38 the authors state: “there were minor differences in activation of the ANF promoter”, however the results show no differences at all, please amend.

-Not standard abbreviations should be listed in alphabetical order.

-Material and methods, lines 176-177: the descriptions of the young-adult and aged rats seem to be inverted, please amend.

-Discussion, lines 443-444: the sentence “Our results support this speculation by (Fig5).” Is incorrect, please amend.

-In some instances the SNP is referred to as p.Arg264Lys in some others as R264K, to avoid confusion a unique acronym should be used throughout the manuscript.

-In fig.2 caption all the transcripts are referred to as mRNAs, however also non coding RNAs are included in the image, please amend.

6. PLOS authors have the option to publish the peer review history of their article (what does this mean?). If published, this will include your full peer review and any attached files.

Reviewer #1: No

Reviewer #2: No

---

## [Author Response · Author response to Decision Letter 0]

25 Feb 2020

Responses to the comments of Editor

Response:

We have ensured our manuscript had met PLOS ONE’s style.

2. Please modify the title to ensure that it is meeting PLOS’guidelines (https://journals.plos.org/plosone/s/submission-guidelines#loc-title ). In particular, the title should be "specific, descriptive, concise, and comprehensible to readers outside the field" and in this case it is not informative and specific about your study's scope and methodology. When amending the title please amend this both on the online submission form (via Edit Submission) and in the manuscript so that they are identical.

Response:

We have changed the title to “TBX5 R264K acts as a modifier to develop dilated cardiomyopathy in mice independently of T-box pathway”, as reviewers also had suggested. 

3. We noticed you have some minor occurrence of overlapping text with the following previous publication(s), which needs to be addressed: https://academic.oup.com/eurheartj/article/29/2/270/433902. 

The text that needs to be addressed involves the Discussion. In your revision ensure you cite all your sources (including your own works), and quote or rephrase any duplicated text outside the methods section. Further consideration is dependent on these concerns being addressed.

Response:

We have confirmed the cited text and described all references without exception. And we corrected the manuscript without overlapping phrase.

4. PLOS ONE now requires that authors provide the original uncropped and unadjusted images underlying all blot or gel results reported in a submission’s figures or Supporting Information files. This policy and the journal’s other requirements for blot/gel reporting and figure preparation are described in detail at https://journals.plos.org/plosone/s/figures#loc-blot-and-gel-reporting-requirements and https://journals.plos.org/plosone/s/figures#loc-preparing-figures-from-image-files. When you submit your revised manuscript, please ensure that your figures adhere fully to these guidelines and provide the original underlying images for all blot or gel data reported in your submission. See the following link for instructions on providing the original image data: https://journals.plos.org/plosone/s/figures#loc-original-images-for-blots-and-gels. In your cover letter, please note whether your blot/gel image data are in Supporting Information or posted at a public data repository, provide the repository URL if relevant, and provide specific details as to which raw blot/gel images, if any, are not available. Email us at plosone@plos.org if you have any questions.

Response:

We have involved all data and images associated with manuscript in Supporting Information named S File, S1-5 Fig. and S1-8 Table.

Response:

We have deleted “data not shown” and added all result data in Supporting Information. About Supporting Information, please see point 4 above. 

 

Responses to the comments of Reviewer #1

Reviewer #1: Authors have performed a very laborious work, considering the characterization of a TBX5 SNP in a transgenic murine model, following a population study. The current manuscript is performed by the application of state-of-art techniques, that efficiently support the data. Some minor and major comments should be considered for the improvement of the manuscript in favour of the authors and the readers. 

Thank you for your kind review and helpful comments to our manuscript.

These are point-by-point response to the reviewers' comments.

Major Comments

1. Authors state to have used transgenic mice of 6-9 months old as an aging model. However, according the literature, mice of that age correspond to human adults around 40 years of age. In order to consider a mouse aged, the minimum age of 18 months should be reached. Authors are advised either to correct the statement, or to present mortality data justifying the selection of the specific age for their murine model.

Response:

According to the standard definition in The Mouse in Biomedical Research 2nd Edition (2007), 6-9 months old mice are between mature adult: 3-6 months and middle age: 10-15 months, so we corrected “aged” to “mature to middle age” throughout this study.

2. Hematoxylin-Eosin staining seems to be poorly performed as nuclei (blue) are not visible in the histology images. Please refine your staining, by improving your H&E protocol and by washing your specimens with PBS for the removal of blood (as visible in Fig. 1A lower panel). Moreover, considering that authors are commenting on intracardiac fibrosis in their transgenic mice, maybe another more specific staining should be applied such as Sirius Red or Masson Trichrome. By the use of these staining techniques the visualization of collagen deposition will be more straight-forward.

Response:

We performed the H&E staining again to select representative sections. We also presented high-magnification photographs for H&E staining. The remarkable deformation such as cell hypertrophy or wavy appearance in cardiomyocytes was not found in mutant mice.

To analyze intracardiac fibrosis, we used Elastica-Masson (EM) staining which is a combination of Verhoeff and Masson trichrome Stain and we use routinely the method for the combined staining of elastic, muscle and connective tissue for histopathology in specimens. Revised representative data are shown in Fig.1A-H.

3. Column graphs are not presented in a sufficient manner. Individual values, showing the distribution and the number of the biological replicates should be present in the graphs (i.e. scatter plotted column graphs) and statistical analysis (i.e. asterisks) should be presented, deducing the statistically significantly differences between the groups.

Response:

A scatter plotted column graph has been created to show the measured values and the number of experimental individuals. We added the analyzed number of individuals (four) in the figure legend. We also added indications of statistically significant differences to the graph. (Fig. 4, 5B and 6) 

4. The use of isoproterenol is not sufficiently justified. Transgenic mice, already present cardiac contractility deficits (e.g. decreased FS%) in the absence of the β-agonist. Therefore, the use of isoproterenol should be better discussed in the Discussion section, considering its transcriptional value in the experiments.

Response:

We used the isoproterenol stimulation test as one of the phenotypic analyzes to evaluate myocardial tolerability in a stress environment. Excessive β-adrenergic receptor stimulation activates the mitogen-activated protein kinase cascade, causing remodeling of myocardial hypertrophy resulting in contraction or diastolic function decreases. There is a literature that focus on the diastolic dysfunction induced by isoproterenol stimulation and analyze specific gene variations [45]. If isoproterenol stimulation causes a significant decrease in cardiac function in the Tbx5R264K/R264K mice group, it is likely that the Tbx5R264K/R264K has caused fragility in cardiomyocytes for some reasons.

We added the significance of use the isoproterenol stimulation test in Discussion section. Please see page 38, lines 560-570.

5. Authors shown that the given SNP in the TBX5 leads to a downregulation of ACTA-1 in the myocardium. Concerning that ACTA-1 is a highly important protein in the cellular cytoskeleton, authors are suggested to perform electron microscopical evaluation of the cardiomyocytes’ structure in the transgenic mice. Therefore, a more detailed representation of the effect of TBX5 SNP on cardiomyocytes’ architecture will be available.

Response:

In our study, Acta1 was upregulated in cardiomyocytes by Tbx5 mutation. We corrected Fig.4 legends to prevent misunderstanding. α-actin has three isoforms: skeletal muscle, cardiac muscle, and smooth muscle. ACTA1, a skeletal actin, and ACTC1, a cardiac actin, bind to myosin in each cell to form sarcomere as the main role. The arcomere is the smallest unit of muscle contraction. Therefore, increased ACTA1 is unlikely to significantly alter the cytoskeleton. Although, observation of the cytoskeleton by electron microscopy as you suggested may be useful to see the detailed morphological changes like a wavy appearance or cell hypertrophy by load on cardiomyocytes, we will examine this in a future research study.

6. If applicable, it would be interesting if the authors could confirm the results from their transgenic murine model in human cardiac biopsies, following simple RT-PCR or Western Blot experiments. Thus, the translational value of the current manuscript will be increased.

Response:

According to the literature, ACTA1 is also increased in human heart failure [39]. Performing a myocardial biopsy from the patient population in this study is not possible for indications and ethics, but it is an interesting option. 

Minor Comments

1. Authors are suggested to change the title from “TBX5 R264K induces chronic heart failure not via T-box pathway” to “TBX5 R264K induces chronic cardiomyopathy independently of T-box pathway”, so that it is more appealing to the reader.

Response:

We changed “chronic heart failure” in the manuscript to “dilated cardiomyopathy (DCM)”, since DCM is defined a chronic disease presenting left ventricular dilatation and systolic dysfunction, except for secondary reasons such as hypertension, valvular disease and coronary artery disease [31]. In our study, the young adult mutant mice developed systolic dysfunction, cardiac dilatation and wall thinning mimicking DCM without secondary reason. So, as reviewer 2 also pointed out, the title was changed to be more impressive as follows: “TBX5 R264K acts as a modifier to develop dilated cardiomyopathy in mice independently of T-box pathway”.

2. Please avoid the use of the term heart failure in your in vivo experiments, as a NYHA scoring should be performed to claim it. Please change the term to cardiomyopathy, when referring to the murine models.

Response:

As your comment, we changed the use of the term from “heart failure” to “cardiomyopathy” in mice.

3. Table 2B: Please correct Ds to LVDs.

Response:

As noted, the notation has been changed from Ds to LVDs.

4. Figure 5, considering the Western Blot analysis of ACTA1 is not self explanatory. Proper labelling of the WB images is missing, as well as the molecular weight of the identified targets and the labelling of the samples. The authors are suggested to change it accordingly.

Response:

We corrected the WB images adding molecular weight markers and sample numbers as shown in Fig. 5A, and created a new column as Fig. 5B showing the change in protein amount.

5. The resolution of the figures is poor, which creates an inconvenience for the reader. Authors are suggested to provide a higher image resolution to the given figures. Moreover, concerning the column plots, authors are suggested to use another software, rather than Microsoft Office Excel, in order to export more appealing plots and figures.

Response:

We have created new columns using different software to make it easier for readers to see.

 

Responses to the comments of Reviewer #2

Reviewer #2: The manuscript describes a new variant of the TBX5 gene located outside the T-box regulatory domain. The variant is over-represented in cardiomyopathy patients and leads to compensated dilated cardiomyopathy in a knock-in animal model. The study is well conducted and organized. Although the new SNP does not seem to be sufficient to induce de-compensated heart disease by itself, the results indicate that it may work as an important phenotype modifier during the evolution of adverse cardiac remodeling in patients. However, the following points must be addressed. 

Thank you for undertaking a review of our manuscript.

These are point-by-point response to the reviewers' comments.

Major points

-The authors focused part of the study on the contribution of differentially expressed mRNAs in the observed phenotypes, have the authors performed a functional enrichment analysis of the differentially expressed transcripts to explore putative underlying biological processes?

Response:

As you indicated, we performed functional enrichment analysis to explore putative underlying biological process, using g:Profiler, a web server for functional enrichment analysis and conversions of gene lists [15]. After 36 genes extracted by microarray were applied to g:Profiler, noncoding genes were excluded, and as a result further analyses were limited to 8 coding genes, all of which were up-regulated genes in mutant mice (Table 4). Of the Biological Processes identified, there were two terms related to myocardium; response to muscle stretch and response to mechanical stimulus (S8 table): this limited the relevant genes to Acta1, Myot, Ankrd1 and Ankrd23. Real time RT-PCR was performed these 8 genes, however no significant changes were observed except for Acta1. Although the relationship between real time RT-PCR results and functional enrichment analysis could not be determined, it is presumed that the number of candidate genes was too small to investigate biological process.

According to your suggestion, we added above considerations in Method section on page 16, lines 246-249, in Result section on page 29, lines 419-424 and in Discussion section on pages 36-37, lines 535-541.

-The majority of the differentially expressed transcripts are still uncharacterized non coding RNAs. To foster future investigation, it should be interesting if the authors could briefly discuss their putative role in the observed phenotype.

Response:

In general, the function of noncoding RNAs has not been fully elucidated. However, as there is a study that have performed noncoding RNA profiles in DCM patients [33], our remaining extracted noncoding RNAs may also have important implications for cardiomyopathy. This will be a future research topic.

We added this consideration in Discussion section on page 37, lines 541-544.

-According to the results shown on page 13 lines 362-369, the differential expression of Myot between wt and mutant mice was not confirmed by PCR. However the authors seem to disregard this finding when discussing the putative role of Myot in disease evolution. Please clarify.

Response:

We performed real time RT-PCR of Myot. However, no significant change was observed as shown in the S1 Fig. Because real time RT-PCR showed no significant data, we deleted the discussion about the role of Myot.

-Why only 2 out of 8 differentially expressed mRNAs were validated by rt-PCR? To confirm the microarray data all the 8 differentially expressed mRNAs must be validated by real time PCR.

Response:

We investigated each using real time RT-PCR. As a result, all except for Acta1 showed no differences. The results are included in S1 Fig. In addition, in the Results section (pages 31-32, lines 454-458), we also performed real time RT-PCR on other sarcomere genes and genes involved in Ca2+ kinetics in cardiomyocites, which could be directly involved in contractile function. However, they showed no significant differences in both groups (S4 Fig.).

-In some instances the authors report the findings as data not shown (see results lines 330, 334, 368). Although these results refer to not statistically significant differences, all the data must be fully available for the reader. Please amend. 

Response:

According to your suggestion, all data were listed in the Supporting Information.

-The authors observed a milder phenotype in the aged mutants with respect to the young-adult ones. How do they explain this finding? 

Response:

As the mice aged, the deposition of fibrosis increased in the endocardium and interstitium. The reactive fibrosis occurs with angiogenesis against the load on cardiomyocytes, and the replacement fibrosis occurs to fill the gap of cardiomyocytes when the cells fall into ischemia or inflammation leading to necrosis [31]. These fibrosis events were observed in mature to middle age Tbx5R264K/R264K mice. The slow progression of these histological changes is dependent on time with increasing contractile dysfunction, according to a mouse model of severe DCM [32]. Thus, the lack of changes, except for systolic dysfunction on echocardiography, in mature to middle aged Tbx5R264K/R264K mice was due to myocardial remodeling that occurred slowly as the mice aged.

According to your suggestion, we added above considerations in Discussion section on page 36, lines 526-534.

- In the light of the limitations acknowledged by the authors themselves (see discussion 457-462), the title should be rephrased given more emphasis to the role of the mutation as a phenotype modifier rather that as an indipendent pathogenic mechanism. 

Response:

As reviewer 1 also pointed, we changed the title to “TBX5 R264K acts as a modifier to develop dilated cardiomyopathy in mice independently of T-box pathway”

-The quantification of the histological results shown in fig 1 is lacking and must be added.

Response:

It is reported that myocardial fibrosis progresses in two patterns, reactive fibrosis and replacement fibrosis. According to the paper by Segura AM et al. (2011), semi-quantification of myocardial fibrosis and computer quantification were highly correlated. So, we used the semi-quantification method and graded the degree of fibrosis (S2 Table) and described in Result section (Table 3). 

According to your suggestion, we added above considerations in Method section on pages 14-15, lines 212-222 and Result section on page 27, lines 384-388.

-The quantification of the WB data shown in figure 5 is lacking and must be added.

Response:

We added a graph quantifying the results of the Western blotting as Fig. 5B.

-The plot shown in figure 6 is reported without indication of the statistical analysis. Although evident, the statistically significant differences must be indicated. 

Response:

We added the indications in fig.6 and other figures requiring significant difference indication.

Minor points

-Abstract, line 38 the authors state: “there were minor differences in activation of the ANF promoter”, however the results show no differences at all, please amend. 

Response:

We corrected that “there was no difference in activation of the ANF promoter”.

-Not standard abbreviations should be listed in alphabetical order.

Response:

We added abbreviations in alphabetical order.

-Material and methods, lines 176-177: the descriptions of the young-adult and aged rats seem to be inverted, please amend.

Response:

We changed the description as requested.

-Discussion, lines 443-444: the sentence “Our results support this speculation by (Fig5).” Is incorrect, please amend.

Response:

We deleted the sentence as suggested.

-In some instances the SNP is referred to as p.Arg264Lys in some others as R264K, to avoid confusion a unique acronym should be used throughout the manuscript.

Response:

We corrected that p.Arg264Lys is unified to R264K throughout the text.

-In fig.2 caption all the transcripts are referred to as mRNAs, however also non coding RNAs are included in the image, please amend.

Response:

We changed the description to “RNAs” because extracted genes by microarray contain various kind of RNAs such as noncoding genes.

---

## [Decision Letter · Decision Letter 1]

6 Mar 2020

TBX5 R264K acts as a modifier to develop dilated cardiomyopathy in mice independently of T-box pathway

PONE-D-19-34630R1

Dear Dr. Hirono,

We are pleased to inform you that your manuscript has been judged scientifically suitable for publication and will be formally accepted for publication once it complies with all outstanding technical requirements.

With kind regards,

Vincenzo Lionetti, M.D., PhD

Academic Editor

PLOS ONE

Additional Editor Comments (optional):

Reviewers' comments:

Reviewer's Responses to Questions

**Comments to the Author**

1. If the authors have adequately addressed your comments raised in a previous round of review and you feel that this manuscript is now acceptable for publication, you may indicate that here to bypass the “Comments to the Author” section, enter your conflict of interest statement in the “Confidential to Editor” section, and submit your "Accept" recommendation.

Reviewer #1: All comments have been addressed

Reviewer #2: All comments have been addressed

2. Is the manuscript technically sound, and do the data support the conclusions?

Reviewer #1: Yes

Reviewer #2: Yes

3. Has the statistical analysis been performed appropriately and rigorously? 

Reviewer #1: Yes

Reviewer #2: Yes

4. Have the authors made all data underlying the findings in their manuscript fully available?

Reviewer #1: Yes

Reviewer #2: Yes

5. Is the manuscript presented in an intelligible fashion and written in standard English?

Reviewer #1: Yes

Reviewer #2: Yes

6. Review Comments to the Author

Reviewer #1: The authors have provided a satisfactory reply to my comments and the manuscript appears to be quite interesting for the scope of the journal.

Reviewer #2: The authors have adequately addressed the comments raised in my previous review and I feel that the manuscript is now acceptable for publication.

7. PLOS authors have the option to publish the peer review history of their article (what does this mean?). If published, this will include your full peer review and any attached files.

Reviewer #1: No

Reviewer #2: No

---

## [Editor Report · Acceptance letter]

10 Mar 2020

PONE-D-19-34630R1 

*TBX5* R264K acts as a modifier to develop dilated cardiomyopathy in mice independently of T-box pathway

Dear Dr. Hirono:

I am pleased to inform you that your manuscript has been deemed suitable for publication in PLOS ONE. Congratulations! Your manuscript is now with our production department. 

With kind regards,

on behalf of

Prof. Vincenzo Lionetti 

Academic Editor

PLOS ONE